# Integrating Indoor Hibernation into the Italian Outdoor Snail Farming System: A Potential Solution for Colder Climates

**DOI:** 10.3390/ani15070914

**Published:** 2025-03-22

**Authors:** Ramona Ștef, Dan Manea, Anișoara Aurelia Ienciu, Emilian Onișan, Dragoș Vasile Nica, Alin Cărăbeț

**Affiliations:** 1Faculty of Agriculture, University of Life Sciences “King Mihai I” from Timișoara, Calea Aradului No. 119, 300645 Timișoara, Romania; ramonastef@usvt.ro (R.Ș.); dan_manea@usvt.ro (D.M.); anisoara_ienciu@usvt.ro (A.A.I.); alincarabet@usvt.ro (A.C.); 2Faculty of Engineering and Applied Technologies, University of Life Sciences “King Mihai I” from Timișoara, Calea Aradului No. 119, 300645 Timișoara, Romania; emilian.onisan@usvt.ro; 3The National Institute of Research—Development for Machines and Installations Designed for Agriculture and Food Industry, Bulevardul Ion Ionescu de la Brad 6, 077190 București, Romania; 4Research Center for Pharmaco-Toxicological Evaluations, Faculty of Pharmacy, “Victor Babeș” University of Medicine and Pharmacy Timișoara, Eftimie Murgu Square No. 2, 300041 Timișoara, Romania

**Keywords:** heliciculture, *Cornu aspersum*, hibernation, outdoor rearing, snail farming management, survival rate, overwintering

## Abstract

Farms operating under the Italian outdoor snail farming (IOSF) system use the Mediterranean snail *Cornu aspersum* as the primary commercial species. Winter hibernation is a critical phase for productivity; however, outdoor survival rates remain a major challenge in farms located in colder regions. This two-year study aimed to adapt indoor hibernation—a practice typically associated with (semi-)intensive snail farming—to the IOSF system as a strategy to mitigate the high mortality rates associated with the sole use of Lutrasil frost cloth (LFC) for overwintering protection. Semiempirical field experiments were conducted across three commercial farms, testing different hibernation scenarios. Results demonstrated that micro shelters significantly reduced the labor time for snail collection compared to hand-picking. Post-hibernation weight loss varied among farms, with significantly higher losses observed in poorly insulated hibernation spaces, although post-purging values remained relatively uniform. The vast majority of snails survived indoor hibernation, yet inadequate thermal insulation yielded significantly higher mortalities despite comparable hibernation durations. Most deaths were attributed to environmental factors rather than predation. The present findings offer practical guidance for snail breeders in regions with colder winters, supporting the integration of indoor hibernation as a viable alternative in IOSF system farms.

## 1. Introduction

During the first decade of the 21st century, the breeding of edible snails, also known as heliciculture, has experienced rapid growth in the former communist countries of Central and Eastern Europe. For example, Romania alone saw the establishment of several hundred farms using Italian outdoor snail farming (IOSF) technology, often referred to as the “Cherasco Method”, between 2003 and 2006 [1,2]. Such development was the result of a considerable increase in global demand for snail meat [3,4] and the availability of large areas of under-exploited arable land in these regions [2]. A large number of these farms were focused on rearing the brown garden snail *Cornu aspersum* (syn. *Helix aspersa*, *Cantareus aspersus*, or *Cryptomphalus aspersus*). This gastropod is one of the most widely cultivated species for its meat: it is prolific, adapts well to captivity, and exhibits a rapid growth rate, making it suitable for intensive farming [5,6,7].

When nighttime temperatures consistently fall below 5 °C and daytime temperatures start to drop below 10 °C, mature specimens of *C. aspersum* begin to prepare for hibernation [6,8,9]. This state of suspended animation helps terrestrial gastropods cope with low temperatures and reduced food availability during winter, enabling them to conserve energy and survive until more favorable conditions return in the spring [10]. From an economic point of view, hibernation is important for maximizing productivity in commercial snail farms since it promotes reproductive success and fecundity via multiple routes (e.g., synchronization of reproductive cycles with favorable environmental conditions or initiation of spermatogenesis) [11,12,13]. During hibernation, glycogen and fat stores are actively consumed, causing inevitable weight loss, whereas moisture loss is greatly reduced [8,14]. Although this decrease in the body weight of hibernating *C. aspersum* snails is a relatively well-studied topic [2,8,14,15,16], there is no information regarding the evolution of this parameter across different subphases of hibernation, that is purging and overwintering [1]. The former term defines the physiological process by which gastropods empty their digestive tract prior to entering hibernation. This preparatory phase reduces the risk of carrying toxins or pathogens inside their bodies while being in a dormant state and ensures that their digestive system is inactive during overwintering [17]. The latter term refers to the period of time when a gastropod stays dormant during the winter months. During this phase, their metabolic rate and activity decrease to conserve energy [2,8,18].

The IOSF system involves raising *C. aspersum* in free-range pens on open pastures with fresh vegetables, thus providing a natural rearing environment [1,3,4]. Within the framework of this farming model, snails typically hibernate outdoors, protected by a thin sheet of Lutrasil frost cloth (LFC) [2,4]. A critical issue for using this approach in regions with colder winters than those of the native areas of this species (e.g., Central and Eastern Europe vs. the Mediterranean) are the elevated mortalities that occur during the winter months [2,6]. A previously developed approach involves the use of the “sandwich” system—structuresoil/LFC/straw/10-centimeter air cushion/high-density polyethylene—in conjunction with ridge-tile/wood micro shelters [2,6]. This approach, used for the first time during the mild winter of 2005/2006, yielded post-hibernation survival rates above those found in natural environments, that is 67–71% vs. 20–50% [8,19], but below those reported for indoor hibernation, which is usually above 80% [16,20,21]. One of the major drawbacks of indoor hibernation is increased energy consumption and associated costs related to purchasing a cold chamber and maintaining an appropriate indoor environment [8]. This can be a problem when using the IOSF system, since its profitability is sensitive to labor and equipment costs [22,23,24]. On the other hand, the only solution left in the case of the low survival of mature *C. asperum* specimens hibernating outdoors involves purchasing new individuals to repopulate the farm [2].

In this context, we aimed to evaluate the feasibility of adapting indoor hibernation technology to the IOSF system. The parameters monitored were body weight loss and survival during the two key phases of hibernation (purging, overwintering), as well as the potential factors underlying post-hibernation mortalities. The efficiency of using micro shelters vs. hand-picking for snail collection was also assessed. Results were obtained from experiments conducted on three snail farms in two consecutive years. The findings of the present study are important since they provide scientists with new insights into the physiological adaptations of snails to hibernation, and provide snail breeders with valuable information for adjusting their farm management.

## 2. Materials and Methods

### 2.1. Study Sites

The experiments were conducted between October 2006 and April 2007 on three snail farms located in Floreşti (farm F1; Mehedinţi county, latitude 44°46′ N, longitude 22°55′ E), Sântuhalm (farm F2; Hunedoara county; latitude 45°51′ N, longitude 22°57′ E), and Ezeriş (farm F3; Caraş-Severin county; latitude 45°24′ N; longitude 21°53′ E). The farms were populated in April 2006 with sexually mature specimens of *Cornu aspersum*, aged 9–12 months. The number of mature snails used as breeding stock for each farm was 7598 specimens for farm F1 (average weight: 10.23 ± 2.82 g), 9750 specimens for farm F2 (average weight: 9.79 ± 3.47 g), and 3152 snails for farm F3 (average weight: 10.745 ± 2.12 g). These snails were purchased from the International Snail Farming Institute (Cherasco, Italy). In accordance with the standard workflow of the IOSF system, the surviving gastropods were transferred into newly established breeding pens three months post-introduction (from June to September 2006), while the juveniles remained in the first established breeding pens [22].

### 2.2. Snail Gathering

The traditional and commonly used method for collecting snails is hand-picking—a time-consuming approach [21,25]. These mollusks are sensitive creatures, thus handling them with care is essential to avoid stress or injury during this process [8,16]. Our previous research revealed that adult specimens of *C. aspersum* preparing for hibernation display a strong preference for wood and ridge-tile micro shelters [6]. As a result, we considered that providing structures or surfaces for snails to adhere to could reduce the time required (in terms of work hours) for these activities.

To test this hypothesis, we ran experiments in October 2006 in the F1 farm. This farm was used as the study site because it provided the best control over experimental conditions, ensuring reliable and consistent data collection. Given the limited resources available in terms of time, labor, and funding, it was also unfeasible to replicate the activity to the other two snail farms. Moreover, our objective was to evaluate the impact of the activity itself, not to compare results between farms.

Briefly, snail gathering was done by hand-picking in three breeding pens, and from micro shelters in another three enclosures. Each breeding pen was 35 m × 2.5 m. The pens where snails were collected by hand-picking did not contain micro shelters. Ridge tiles, wooden pallets, rubber sheets, and pieces of cardboard, with a surface of up to 0.25 m^2^, were used as micro shelters in the other enclosures. Hand-picking was used in these pens only in exceptional cases, such as a small number of snails failing to attach to the micro shelters or unfavorable weather conditions (imminent freezing) demanding quick collection. Snail gathering started after the vegetation inside the pens was cut to a height of 10 to 20 cm—as recommended by the IOSF system [4]. The same two individuals collected the snails to ensure consistency in harvesting efficiency and reduce variability due to differences in worker experience or technique.

### 2.3. Snail Hibernation

Ideally, hibernation of *C. aspersum* takes place in a cold room at a temperature between 4 °C and 7 °C [20]. The purchase/use of such equipment is, however, not cost effective when the number of snails is small to moderate (up to 15,000 specimens). The same principle applies to contexts with limited financial resources, such as in the case of the IOSF system—which relies on smaller investments compared to mixed and intensive rearing systems [2,4,20]. As a result, each of the aforementioned farms used a semiempirical approach for snail hibernation. This approach was developed based on theoretical knowledge from specialty literature; practical data collected from foreign farmers (e.g., Italy, France); and available material, equipment, and financial resources.

After being collected from the pens, only live specimens were retained for further use, and all dead specimens were discarded. These individuals were dried for one week in purging cages in farms located in Floreşti and Sântuhalm (500 individuals per purging cage); and in micromesh bags in the Ezeriş farm (300 individuals per bag). The purging cages were made of galvanized steel wire mesh on a wooden frame (80 cm × 80 cm × 30 cm). Proper ventilation was provided using an air fan. This purging stage is a key step for ensuring successful overwintering because it enables gastropods to empty their intestines and reduce water body percentage [8]. Prior to hibernation, all dead snails were removed from the purging cages (micromesh bags), which were then manually cleaned of feces and other debris using a plastic brush.

To monitor the hibernation process, 15 mature snails were randomly selected per location. We aimed to test a new hibernation method, and as a result, this relatively small sample size was deliberately chosen to minimize risk and resource use while still providing valuable insights. These snails were individually marked using black acrylic paint to ensure easy identification throughout the study. The markings allowed researchers to track their survival status during the hibernation period. Their height was measured prior to purging using digital calipers. Their pre-purging, post-purging, and post-hibernation weights were determined using a jewelry balance with a precision of 0.01 g.

The overwintering conditions were as follows: (*i*) Floreşti: hibernation in purging cages (500 individuals per cage), placed in an underground cellar (below ground) with 10 cm-thick expanded polystyrene (ESP) insulation. (*ii*) Sântuhalm: similar purging cages, placed in a non-insulated basement illuminated by daylight (above-ground, with windows); (*iii*) Ezeriş: potato bags (300 individuals per bag) placed in perforated plastic crates (80 cm × 50 cm × 20 cm) in an ESP-insulated half-buried root cellar. When needed, an air fan was used to promote ventilation.

At the F1 farm, snails hibernated in a completely dark environment. At the F2 farm, by contrast, mature specimens hibernated under a natural photoperiod. This was due to the constructive features of the hibernation space, which allowed daylight exposure. At the F3 farm, hibernation occurred under a regulated photoperiod, with timers being used to simulate the desired cycles. Thermohygrometers were used to measure both the indoor temperature and relative humidity. These parameters were generally measured weekly, when environmental conditions were stable. More frequent checks (twice a week or daily) were conducted when indoor/outdoor conditions (temperature/humidity) fluctuated significantly and before the snails emerged from hibernation. Hibernation densities varied between 2600 specimens/m^3^ (farms F1 and F2) and 3750 snails/m^3^ (farm F3).

The snails were awakened from hibernation in March–April 2007, depending on the location and environmental conditions. The overwintering structures were watered daily for three consecutive days to promote spring arousal. The overwintering success was assessed based on the survival rate. We assigned a probable death cause to all shells, as previously described [6,17].

### 2.4. Statistical Analysis

The snails were first grouped into specimens collected from pens with and without micro shelters (hand-picking only). Intergroup differences in the number of specimens used to populate these enclosures in the spring were determined using a *t* test. A similar approach was applied to mature gastropods gathered in fall to be transferred from the breeding pens into the hibernation spaces. The duration of activities related to snail harvesting was measured for both collection methods. A *t* test was also conducted to identify the most time-efficient method for collecting mature *Cornu aspersum* snails. All these datasets were first checked for normality and homoscedasticity using, respectively, a Shapiro–Wilk test and an F test.

Differences in shell height were analyzed using one-way ANOVAs, provided that these data displayed normality (verified via Shapiro–Wilk tests) and homogeneity of variance (tested with a Levene’s test). In case of significant differences, post hoc analysis was conducted using the Newman–Keuls procedure [26]. The same methodology was used for the pre-purging body weight, post-purging body weight loss, and post-hibernation body weight loss. Chi^2^ tests were then conducted on 3 × 2 contingency tables to identify differences in post-purging survival and post-hibernation survival among different farms. For significant differences, post hoc testing was run for all paired comparisons using Chi^2^ tests based on 2 × 2 contingency tables [27]. A similar approach was used for the distribution of shell condition classes [6]. All statistical analyses were performed using Statistica version 8 (StatSoft Inc., Tulsa, OK, USA). In all of the cases, a *p* value of less than 0.05 was considered significant. We have adjusted the values of Chi^2^ according to Yates’s correction for continuity to reduce the error in approximation.

## 3. Results

### 3.1. Snail Gathering

The number of snails introduced in each breeding pen when the F1 farm was established, the number of specimens collected for hibernation, and the work hours needed to gather all the snails from the pens are shown in Table 1. No exceptional weather conditions occurred during snail collection, making hand-picking in pens with micro shelters unnecessary. Datasets were normally distributed (Shapiro–Wilk tests: *p* ≥ 0.265) and homoscedastic (F tests: *p* ≥ 0.607). No significant differences existed between these groups with respect to the first variable (*t* test: *t* (4) = 0.172, *p* = 0.871) and the second variable (*t* test: *t* (4) = 0.147, *p* = 0.890). Compared to hand-picking only (Figure 1a), the use of micro shelters (Figure 1b–d) involved significantly fewer working hours (*t* test: *t* (4) = 3.501, *p* = 0.025; 25.33 ± 2.52 vs. 17.33 ± 3.06).

### 3.2. Body Weight Loss

The measured values for shell height and pre-purging body weight in marked snails are given in Table 2. Datasets for both variables were normally distributed (Shapiro–Wilk tests: *p* ≥ 0.256) and homoscedastic (Levene’s test: *p* ≥ 0.585). A one-way ANOVA revealed no difference in shell height (ANOVA: F (2, 42) = 0.74, *p* = 0.297). Similar results were obtained for pre-purging body weight (F (2,42) = 0.65, *p* = 0.527). These data attest to the homogeneity of experimental animals in terms of age and size.

The mean percentages of weight loss during the purging period were 14.67% for the F1 farm, 18.37% for the F2 farm, and 15.41% for the F3 farm. The corresponding datasets for weight loss (expressed in grams), which are shown in Figure 2a, showed a normal distribution (*p* ≥ 0.439) and equality of variance between groups (*p* = 0.531). Application of a one-way analysis of variance showed no significant differences among the groups investigated (ANOVA: F (2, 87) = 0.59, *p* = 0.555).

The mean percentages of weight loss during hibernation were 13.03% for the F1 farm, 19.12% for the F2 farm, and 12.64% for the F3 farm. Absolute datasets for weight loss (expressed in grams), which are given in Figure 2b, were normally distributed (*p* ≥ 0.354) and homoscedastic (*p* = 0.062). It was found that post-hibernation weight loss in marked gastropods was different among groups, as revealed by the results of the ANOVA (F (2, 42) = 4.20, *p* = 0.025). Weight loss at the end of hibernation was significantly higher at the F2 farm compared to both the F1 farm and the F3 farm (Figure 1b; Newman–Keuls tests: *p* ≤ 0.035). However, no significant differences existed between the F1 farm and the F3 farm (Figure 1b; Newman–Keuls test: *p* = 0.857).

### 3.3. Overwintering Survival

Data related to microenvironment parameters (temperature, relative humidity, photoperiod) in overwintering spaces and survival at different time points for the three snail farms are given in Table 3. Mortalities during the purging period were very low, with no significant differences observed among the farms investigated (Chi-square test: χ^2^ = 0.21, *p* = 0.901). In addition, no mortalities occurred for the marked specimens during this period of time.

The F1 farm and the F3 farm showed much lower variations in temperature and relative humidity in the hibernation spaces, suggesting a more stable overwintering microclimate. No major incidents were recorded until February 2007, when external temperatures at the F2 farm increased, accompanied by a rise in humidity. This phenomenon, easily detectable by the formation of condensation on the walls of the storage area, led to the premature emergence from hibernation of a subset of the snail population. Intervention was carried out by manually eliminating the dead individuals, and alive, awake specimens were re-purged to ensure the proper elimination of residual gut contents. The survival rates were roughly in the range of 70–80%; the highest values were found for the F1 farm and F3 farm, both about 10% above those seen in the F2 farm (Table 3).

Statistical analysis revealed significant inter-group differences with respect to hibernation success (Chi-square test: χ^2^ = 124.37, *p* < 0.001). Survival was significantly higher for the snails on the F1 farm compared to those on the F2 farm (Chi-square test: χ^2^ = 93.62, *p* < 0.001). Similar results were also obtained for the F3 farm (Chi-square test: χ^2^ = 60.61, *p* < 0.001). However, no significant differences existed between the F1 and F3 farms (Chi-square test: χ^2^ = 1.12, *p* = 0.289). Regarding the marked specimens of *C. aspersum* snails, overwintering mortalities were similar; more precisely, two specimens from the F1 farm (13%), three specimens from the F2 farm (20%), and two specimens from the F3 farm (13%) died.

### 3.4. Mortality Factors

A probable cause of death was assigned to all snails based on the post-hibernation condition of their shells. Table 4 shows the distribution of these strata. Significant differences were observed in the distribution of shell condition classes across dead snails (Chi-square test: χ^2^ = 702.86, *p* < 0.001). The frequency of intact shells was the highest for the F2 farm, but was similar for the F1 farm and the F3 farm (Table 4). The proportion of smashed shells was much greater for the F1 farm relative to both the F2 farm and F3 farm (Table 4). In addition, the F1 farm displayed, by far, the highest incidence of shells with small holes (Table 4).

## 4. Discussion

This study brings several new insights into the field of heliciculture, particularly in the context of snail hibernation management. While previous studies focused on natural or outdoor hibernation [2,5,6,8,16,28,29], this research assesses the feasibility of integrating indoor hibernation within the IOSF model, balancing survival rates with economic constraints. The present investigation also takes a different approach by focusing on the purging phase and overwintering phase as specific stages of hibernation, rather than examining weight loss across the entire hibernation period [2,5,15,16,30,31]. Moreover, our results demonstrate that non-insulated environments are associated with greater weight loss and higher mortalities post-hibernation. Finally, this study shows that the use of micro shelters vs. hand-picking in snail gathering reduces labor hours, which is crucial for farm management efficiency. These findings contribute to the optimization of snail farming practices, offering practical recommendations for farmers in colder climates and advancing the scientific understanding of *Cornu aspersum* hibernation physiology.

### 4.1. Snail Gathering

The number of snails introduced (recovered) from the breeding pens was similar, irrespective of the use of micro shelters. This demonstrates that the farm population was managed as recommended by the IOSF system; breeding pens of comparable sizes should be populated with a similar number of mature *C. aspersum* snails [4,22]. These results also show that the method of collection did not significantly affect the number of gastropods collected. We also note that a small percentage of the breeding herd appeared to be missing in the fall, when the snails were collected for indoor hibernation. Several potential factors, including natural predation, snail migration or movement, diseases, environmental factors (e.g., drought, lack of proper shelter, sudden change in soil pH, inadequate food supply), and adaptation-related mortalities, may help account for these findings [2,4,5,7,19,21]. However, the collection methods used involved different durations, with the use of micro shelters significantly reducing work time compared to hand-picking only. The use of micro shelters can thus streamline this process by reducing the amount of time and labor required for snail collection from breeding pens.

### 4.2. Body Weight Loss

The lack of significant inter-group differences in post-purging body weight loss indicates that purging was conducted under optimum conditions on all the analyzed farms. However, post-hibernation weight loss was significantly higher for the F2 farm. Published data indicate that substantial weight loss adversely impacts gastropod overwinter survival [2,5,6,19]. Higher post-hibernation mortality rates should hence be expected at this farm (see below). The relative loss of weight observed in this study during hibernation (13–19%) aligns with field data on snail farming. For example, Dupont-Nivet et al. reported a mean percentage of post-hibernation weight loss of 21.4% for three successive generations of mature *C. aspersum* snails reared in commercial heliciculture farms [14]. Drăghici et al. found weight loss between 16.33% and 20.51% during indoor hibernation of juvenile specimens of the same species [8]. Research conducted by Şereflişan and Duysak also showed a weight reduction of 18–22% during hibernation in *Helix pomacella* (Mousson, 1854), *Eobania vermiculata* (Müller, 1774), *Helix melanostoma* (Draparnaud, 1801), and *Helix asemnis* (Bourguignat, 1860) [30]. These findings lend support for a similar pattern of weight loss over winter across gastropods of different ages and species.

### 4.3. Overwintering Survival

Within the framework of the IOSF system, sexually mature snails acquired at farm population as breeding stock are typically sold in the fall of the same year [1,2,4,22,31,32,33]. If the remaining specimens exhibit elevated mortalities over winter, the farm can suffer economic losses due to disrupted breeding cycles, reduced offspring, and elevated repopulation costs. This negatively impacts farm productivity and profitability prospects [2,21]. The present results demonstrate that hibernation under (semi-)controlled indoor conditions enables the vast majority of mature specimens of the brown garden snail *C. aspersum* to successfully survive over winter. The measured survival rates are consistent with those reported for the mixed system, i.e., 75–85% [20,34,35]. However, our approach requires less investment; for example, it uses available spaces (cellars, basements) and upgrades with minimal costs instead of requiring the purchase of a cold chamber.

It is likely that the significantly elevated post-hibernation mortalities and weight loss observed on the F2 farm emerged, at least partly, from the use of a non-insulated, above-ground basement as a hibernation chamber. In contrast, the F1 and F3 farms utilized EPS-insulated, buried or half-buried cellars. Given its closed-cell structure, composed of 98% air, EPS is a good thermal insulator [36]. In addition, the soil has the ability to retain heat and regulate temperature, acting as a natural insulator [37]. These data suggest that indoor hibernation in spaces lacking proper thermal insulation adversely affects the overwintering survival of adult *C. aspersum* snails. Temperature fluctuations inside the hibernation space that deviate from the optimal range for dormancy are a potential driver of this outcome. When temperatures exceed 5–8 °C, premature awakening can occur—as observed in the F2 farm—leading to increased energy expenditure and depletion of stored reserves [8,35]. On the other hand, snail death through freezing, in the case of prolonged exposure to temperatures below 0 °C, is also plausible [2,38]. These findings highlight the importance of maintaining stable and optimal thermal conditions during hibernation to maximize survival outcomes.

Survival rates in the F1 and F3 farms were higher than those seen for both outdoor hibernation (protection with LFC as per the IOSF system) [2,28,39,40] and protected outdoor hibernation (using the “sandwich” system) [6]. These results favor the use of indoor overwintering, instead of the latter two approaches, for successful farming of the brown garden snail *Cornu aspersum* in colder climates. Besides the advantages related to proper control of temperature, humidity, and other environmental factors (e.g., predators), indoor hibernation also allows for closer monitoring and care of the snails. Thus, their condition can be regularly checked, and any issues can be addressed promptly [20].

### 4.4. Mortality Factors

Most dead snail shells were intact, irrespective of farm analyzed. The frequency of this shell condition class was above that reported during the same period of time in outdoor snail farms using the “sandwich” system for overwinter protection of *C. aspersum* adults [6]. Although the precise cause of death cannot be established, malacological literature suggests that intact shells typically reflect non-predation deaths (e.g., disease, stress, senescence) [41,42]. From this point of view, indoor hibernation seems more effective than outdoor hibernation in limiting predation-related mortalities in heliciculture farms. Notably, the proportion of intact shells was significantly greater for the F2 farm. Interestingly, mature specimens from this location exhibited the highest post-hibernation death rate and body weight loss. It is likely that starvation related to premature arousal and re-entering hibernation, [2,43] and/or exposure to low temperatures [6,8,9]—as observed in this farm—were major contributors to this outcome (see Section 4.3).

The presence of large holes in the back of the body whorl or in the spire of dead snail shells is generally associated with predation by small mammals, especially rodents (mice, rats) [6,43]. This type of shell lesion is commonly encountered in outdoor snail farms [2,4,6,20,21,44]. The absence of this shell class hence supports the absence of rodents inside the overwintering chambers of the investigated snail farms.

Smashed shells had a relatively low incidence among the three snail farms. This type of injury usually stems from handling snails during harvesting and purging [2], or from bird, insect, or mammal predation [2,6,41,42]. The type of shell damage was not assessed prior to hibernation, but it is extremely improbable that the aforementioned lesions occurred during overwintering since indoor hibernation provides land snails with an optimal protection against predators [20]. It is also reasonable to assume that dead shells with small holes are the result of insect-induced injuries [43,45]. Insects from the *Staphylinide* and *Carabide* are the most frequent predators of land snails [41]. However, other insects, like some *Silphidae* (both adults and larvae) and *Lampyridae* (only larvae) regularly include these mollusks in their diet [42]. Considering the relatively high number of malacophagous predators of terrestrial gastropods [41] and the lack of data on the entomological profiles of these snail farms, it is difficult to derive relevant conclusions about the predator behind this type of shell lesion. Based on available data and the pattern of damage (small circular holes), it can, however, be inferred that among the plausible candidates are *Drilus* beetle larvae—a specialized predator that regularly includes *C. aspersum* in its diet [46].

## 5. Conclusions

This study provides valuable insights into the adaptation of indoor hibernation for *Cornu aspersum* within the Italian outdoor snail farming (IOSF) system, particularly in colder climates. The findings demonstrate that indoor hibernation significantly improves survival rates compared to outdoor methods relying solely on Lutrasil frost cloth (LFC) or the “sandwich” system. However, the study also highlights the critical role of thermal insulation in hibernation chambers, as inadequate insulation leads to higher weight loss and increased mortality. Farms using insulated, underground or semi-buried cellars as hibernation spaces had notably higher survival rates than those relying on non-insulated, above-ground structures for this purpose.

In addition, the use of micro shelters significantly reduced the labor time of snail collection compared to hand-picking, offering a practical improvement for farm management. Predation played a minor role in post-hibernation mortality, with most deaths attributed to environmental factors, such as temperature fluctuations and inadequate hibernation conditions. These findings suggest that indoor hibernation, when properly managed, represents a viable solution for snail farmers in colder climates, balancing economic feasibility and improved survival rates.

## Figures and Tables

**Figure 1 animals-15-00914-f001:**
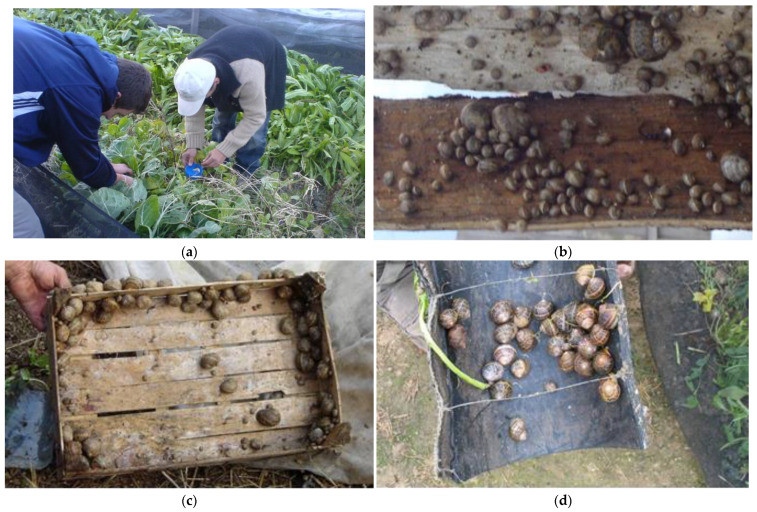
Transferring snails to an indoor environment for hibernation. (**a**) Snail hand-picking; (**b**) adult gastropods attached to the lower surface of timber slabs; (**c**) mature specimens attached on the inner surface of wooden crates; (**d**) rubber micro shelters.

**Figure 2 animals-15-00914-f002:**
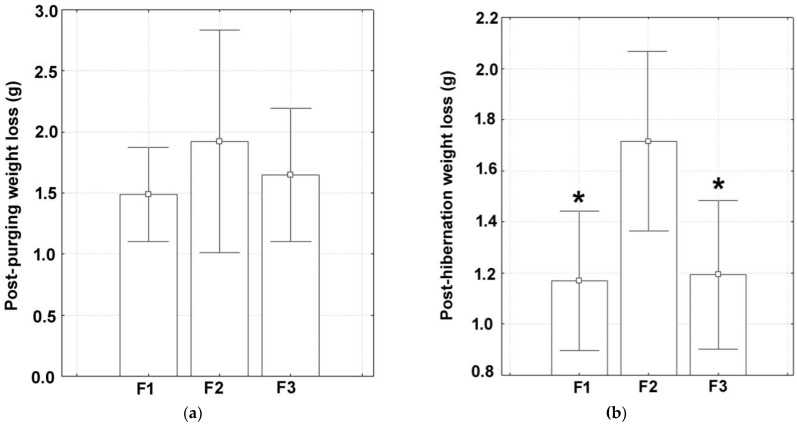
The measured values for (**a**) post-purging weight loss and (**b**) post-hibernation weight loss in the three snail farms investigated. Data are given as means (box) with standard deviations (error bars). Marked boxes (*) indicate significant differences compared to the F2 farm. (Newman–Keuls test, ***—*p* < 0.001, **—*p* < 0.01, *—*p* < 0.05).

**Table 1 animals-15-00914-t001:** Snail population, harvesting efficiency, and work hours in breeding pens.

Breeding Pen	Snail Number		Work Hours
	Population	Harvesting	
BP1 *	1250	1054 (84.32%)	14
BP2 *	1267	1083 (85.48%)	18
BP3 *	1287	1120 (87.02%)	20
BP4	1234	1082 (87.68%)	23
BP5	1289	1104 (85.65%)	28
BP6	1271	1081 (85.05%)	25

BP, breeding pen. Marked breeding pens (*) indicate enclosures in which snails were collected using micro shelters. The number of snails is given as absolute values for the second column and as absolute values with the percentage of snails recovered (versus the population time) in parentheses for the third column.

**Table 2 animals-15-00914-t002:** Shell height and pre-purging body weight of *C. aspersum* at the investigated farms.

F1 Farm		F2 Farm		F3 Farm	
Shell Height	Body Weight	Shell Height	Body Weight	Shell Height	Body Weight
25	8.23	27	9.15	31	11.57
26	10.94	28	9.73	29	12.12
32	11.27	25	8.43	27	12.45
29	8.58	24	11.72	29	10.94
22	11.22	31	11.54	22	12.13
29	10.31	33	9.04	27	9.05
27	10.82	28	12.33	29	12.35
26	11.41	32	12.86	33	9.47
30	9.02	30	10.65	26	8.56
32	8.77	26	8.29	25	11.29
31	10.26	30	10.91	26	11.41
25	11.38	34	9.06	31	10.02
27	8.42	28	10.11	29	10.27
29	11.28	34	12.03	26	9.33
31	10.03	22	11.08	30	9.29
28.07 (2.94)	10.13 (1.20)	28.80 (3.65)	10.46 (1.47)	28.01 (2.80)	10.68 (1.33)

F1 farm, Floreşti (Mehedinţi county); F2 farm, Sântuhalm (Hunedoara county); F3 farm, Ezeriş (Caraş-Severin county). The data in the last row are presented as the mean value with one standard deviation (in parentheses).

**Table 3 animals-15-00914-t003:** Hibernation conditions and survival of *Cornu aspersum* adults in the analyzed farms.

	F1 Farm	F2 Farm	F3 Farm
Temperature (°C)	3 °C (±2 °C)	3 °C (±5 °C)	4 °C (±2 °C)
Photoperiod (dark hours: light hours)	24 h:0 h	natural photoperiod	16 h:8 h
Relative humidity (%)	70% (±5%)	50% (±20%)	65% (±8%)
Number of snails (pre-purging)	6524	8602	2485
Mortalities (post-purging)	78	96	29
Post-purging mortalities (%)	1.20%	1.12%	1.17%
Number of snails alive post-purging	6446	8506	2456
Post-hibernation mortalities	1418	2637	509
Number of snails alive post-hibernation	5106	5965	1976
Overwinter survival	78.26%	69.34%	79.52%
Overwintering duration (days)	116	110	105

F1 farm, Floreşti (Mehedinţi county); F2 farm, Sântuhalm (Hunedoara county); F3 farm, Ezeriş (Caraş-Severin county). Temperature, photoperiod, and relative humidity were measured exclusively within the hibernation chambers, without considering external environmental conditions. Data for temperature and relative humidity are given as a means with standard deviations (in parentheses).

**Table 4 animals-15-00914-t004:** Post-hibernation shell condition of *Cornu aspersum* adults in different farms.

	F1 Farm	F2 Farm	F3 Farm
Intact shell	1023	2522	353
Smashed shell	254	32	34
Small holes	141	83	122

F1 farm, Floreşti (Mehedinţi county); F2 farm, Sântuhalm (Hunedoara county); F3 farm, Ezeriş (Caraş-Severin county).

## Data Availability

All relevant data are within the paper.

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
