# Peer review of "Integrating Indoor Hibernation into the Italian Outdoor Snail Farming System: A Potential Solution for Colder Climates"

_animals, 2025, doi:10.3390/ani15070914_

Round 1

Reviewer 1 Report

Comments and Suggestions for Authors

Dear authors,

Thank you for this very interesting study on heliciculture.

The introduction provides sufficient background and include relevant references.

Also, discussion is well written. Statistical methods are valid and correctly applied.

Below you will find my comments about your paper.

I believe that you must write more details in Materials and Methods:

Line 116 - 119: How many snails (number or Kg) did you used per farm? And how many gastropods survived and transferred into newly established breeding pens (number or percentage)?

Line 130: You mentioned only the F1 farm. What about the others 2 farms mentioned in 2.1 Section? You did not run experiments there?

Line 130 - 133: In the pens that snails collected by hand-picking, there were shelters? If yes you collect also snails from them? Additionally in the pens with micro shelters, you collect shelters from the ground?

Line 156: You select 15 snails per location. You mean 15 per farm? The percentage (15/500) is too low.

Line 162 – 167:  This paragraph needs to be more detailed. Please write about photoperiod, temperature inside the building, humidity levels and snails density.

Section 3.1: You should add results about farm F2 and F3. The same applies in Table 1.

Table 1: Work hours are per person?

Table 2: You should add the mean snail weight per farm

Line 261: I think you mean Table 3.

Table 3: You must correct the Caption.

Table 3: Temperature was taken inside the building?

Table 3: Relative humidity is a mean value? You should add St. dev. In F1 and F3 farm the humidity remains stable all the time?

Table 3: Last 3 Lines (Intact shell, smashed and small holes) can be a separate Table

Line 309: How much time did it take to put the microshelters in the farms? Was it significant?

Comments on the Quality of English Language

Some improvements in the language need to be made (Line 153,154,161). You should also take care of the format and make sure is 100% the Journal’s format.

Author Response

# Reviewer 1

Q1. Line 116 - 119: How many snails (number or Kg) did you used per farm? And how many gastropods survived and transferred into newly established breeding pens (number or percentage)?

R1. We understands the reviewer’s point of view and acknowledge the importance of providing requested data. Fortunately, detailed data on the number and weight of snails used per farm were available, and they have been included in the revised version of the manuscript; please see penultimate sentences from the subsection 2.1. Study Sites:

The number of mature snails used as breeding stock for each farm was: 7,598 specimens for farm F1 (average weight: 10.23 ± 2.82 g), 9,750 specimens for farm F2 (average weight: 9.79 ± 3.47 g), and 3,152 snails for farm F3 (average weight: 10.745 ± 2.12 g).

Unfortunately, we regret that we do not have detailed records on the number or percentage of gastropods that survived and were subsequently transferred into newly established breeding pens. However, this was not within the purpose of the present investigation.

While this limitation does not affect the main conclusions of our study, we acknowledge its relevance and will consider addressing it in future research.

Q2. Line 130: You mentioned only the F1 farm. What about the others 2 farms mentioned in 2.1 Section? You did not run experiments there?

R2. Yes. See details at R6, below.

Q3. Line 130 - 133: In the pens that snails collected by hand-picking, there were shelters? If yes you collect also snails from them? Additionally in the pens with micro shelters, you collect shelters from the ground?

R3. There were no microsheleters in the pens from which snails were collected by hand picking. The snails were collected from the ground in the pens with micro shelters only under exceptional cirmcumstances. These issues are clarified in the revised version of the manuscript; please see the last paragraph from the section subsection 2.2. Snail Gathering; the updated text now reads:

Briefly, snail gathering was done by hand-picking in three breeding pens, and from microshelters in another three enclosures. Each breeding pen was 35 m x 2.5 m. The pens where snails were collected by hand-picking did not contain microshelters. Ridge tiles, wooden pallets, rubber sheets, and pieces of cardboards, with a surface of up to 0.25 m2, were used as microshelters in the other enclosures. Hand-picking was used in these pens only in exceptional cases, such as a small number of snails failing to attach to the mi-croshelters or unfavorable weather conditions (iminent freezing) demanding quick col-lection. Snail gathering started after the vegetation inside the pens was cut to a height of 10 to 20 cm—as recommended by the IOSF technology [4]. The same two individuals collected the snails to ensure consistency in harvesting efficiency and reduce variability due to differences in worker experience or technique..

In addition, no exceptional or adverse weather conditions were observed during the snail collection period, thereby eliminating the  need for manual ground picking in pens equipped with microshelters. This is also mentioned at subsection 3.1. Snail gathering; please see the second sentence from this part.

No exceptional weather conditions occurred during snail collection, making hand-picking in pens with microshelters unnecessary.

Q4. Line 156: You select 15 snails per location. You mean 15 per farm? The percentage (15/500) is too low.

R4. We have indeed selected 15 snails per farm although this sample size represents a small fraction of the total farm population. This approach was used as a pilot-scale approach to monitor hibernation progress. However, we deliberately chose this number. First, we were testing a new method of hibernation. Our aim was therefore to use these 15 snails as a barometer to gather preliminary insights into weight changes and survival under these conditions. At this stage, more extensive sampling was also not feasible due to the time, labor, and resource limitations inherent in monitoring individual snails on a daily or weekly basis. Furthermore, a limited sample size allowed us to quickly detect any significant issues and refine the protocol before undertaking more extensive trials.

A brief summary of these reasons was incorporated into the revised manuscript; please see the second sentence of the third paragraph from the subsection 2.3. Snail Hibernation in the revised from of the manuscript:

We aimed to test a new hibernation method, and as a result, this relatively small sample size was deliberately chosen to minimize risk and resource use while still providing valuable insights. ”

Q5. Line 162 – 167:  This paragraph needs to be more detailed. Please write about photoperiod, temperature inside the building, humidity levels and snails density.

R5.

We understand the reviewer’s point of view and inproved the manuscript accordingly; we included a new paragraph providing the requested details in the revised version of our manuscript; please see the penultimate paragraph from the subsection 2.3. Snail Hibernation:

At the F1 farm, snails hibernated in a completely dark environment. At the F2 farm, by contrast, mature specimens hibernated under a natural photoperiod. This was due to the constructive features of the hibernation space, which allowed daylight exposure. At the F3 farm, hibernation occurred under a regulated photoperiod, with timers being used to simulate the desired cycles. Thermohygrometers were used to measure both indoor temperature and relative humidity. These parameters were generally measured weekly when environmental conditions were stable. More frequent checks (twice a week or daily) were conducted when indoor/outdoor conditions (temperature/humidity) fluctuated significantly and before the snails emerged from hibernation. Hibernation densities varied between 2,600 specimens/m3 (farms F1 and F2) and 3750 snails/m3 (farm F3).

Q6. Section 3.1: You should add results about farm F2 and F3. The same applies in Table 1.

R6. These activities were conducted only in the farm F1. This issue has been addressed in the revised version of the manuscript, including the rationale behing our decision; please see the the second paragraph of the subsection. 2.2. Snail Gathering –correspoding to the section 3.1. Snail gathering:

To test this hypothesis, we run experiments in October 2006 in the F1 farm. This farm was used as the study site because it provided the best control over experimental conditions, ensuring reliable and consistent data collection. Given the limited resources available in term of time, labor, and funding, it was also unfeasable to replicate the activity to the other two snails farms. Moreover, our objective was to evaluate the impact of the activity itself, not to compare results between farms.

Q7. Table 1: Work hours are per person?

R7. No. In the improved version of the mnuscript we mentione that “ The same two individuals collected the snails to ensure consistency in harvesting efficiency and reduce variability due to differences in worker experience or technique.”- the last sentence from the subsection. 2.2. Snail Gathering. So the real work hours per person would be halved.

Q8. Table 2: You should add the mean snail weight per farm

R8. The requested data have been inserted in the revised manuscript; see last row in Table 2.

Q9. Line 261: I think you mean Table 3.

R9. Yes, We have the necessary corrections in the revised form of our manuscript. The updated title now reads; “ Table 3. Hiberation conditions and survival of Cornu aspersum adults in analyzed farms.

Q10. Table 3: You must correct the Caption.

R10. Ok. A suitable title was given to this table in the revised version of the manuscript, i.e., Hibernation conditions and survival of Cornu aspersum adults in analyzed farms.

Q11. Table 3: Temperature was taken inside the building?

R11. Yes. To clarify this issue a new sentence was added at the legend of Table 3; it reads “Temperature, photoperiod, and relative humidity were measured exclusively within the hibernation chambers, without considering external environmental conditions.

Q12. Table 3: Relative humidity is a mean value? You should add St. dev. In F1 and F3 farm the humidity remains stable all the time?

R12. In the revised manuscript, we present the measured values for temperature and relative humidity as mean with one standard deviation, as recommended by the reviewer. This clarification has been also included in the legend of this table. It can be also easily observed from the Table 3 that humidity in F1 and F3 farms was relatively stable compared to F2, as evidenced by their lower standard deviations. These findings were also introduced in the revised version of the manuscript; see the first sentence, second paragrph

The F1 farm and the F3 farm showed much lower variations of temperature and relative humidity in the hibernation spaces, suggesting a more stable overwintering microclimate.”

Q13. Table 3: Last 3 Lines (Intact shell, smashed and small holes) can be a separate Table

R13. We understand the reviewer’s point of view and have made the necessary changes in the revised version of the manuscript; we separated the last 3 lines in a new table, Table 4.

Q14. Line 309: How much time did it take to put the microshelters in the farms? Was it significant?

R14. The time required to put microshelters per breeding pen (35 m × 2.5 m, 20–30 microshelters) ranged between 20 and 45 minutes. This duration was not significant relative to the time needed to gather all the snails from the breeding pens.

Q15. Comments on the Quality of English Language

Some improvements in the language need to be made (Line 153,154,161). You should also take care of the format and make sure is 100% the Journal’s format.

R15. Thank you for your feedback. We have carefully revised the language not only in the specified lines (153, 154, 161), but laso throghout the entire manuscript to improve clarity and precision. We also ensured that the formatting in the revised version of the manuscris fully aligns with the journal’s requirements.

Reviewer 2 Report

Comments and Suggestions for Authors

In my country (cold climate) the semi-intensive system of Cornu aspersum production dominates (on average 30 tonnes/ha). In this system 50 % of field breeding pens are covered with wooden pallets on which dry feed for snails is poured, and the bottom of these pallets is used as microshelters for snails. During the autumn harvest, all sown fodder vegetation is already eaten by snails. Mature snails are therefore easily picking up in autumn and then hibernate in air-conditioned hibernation chambers, with losses of 10-15 %. The research carried out by the authors concerns the technology of extensive farming, without feeding animals,where snails usually winter in their field pens under an appropriate frost-resistens fabric. In this context, the introduction of the described microshelters to the Italian technology with lush vegetation remaining in autumn indeed, as the authors have shown significantly improves picking-up, and wintering indoors contributes to the reduction of individual losses in spring. The work is also correct in formal terms and suitable for publishing after minor corrections below.

Additional comments:

References should be arranged in alphabetical order;

Please, but only if possible, suplement the work with information on the improvement of economic parameters caused by the modification of the tested technology. For example, by how many hours the working time was shortened during picking-up, by how much the material losses were reduced with lower mortality, etc.

Author Response

Q1. In my country (cold climate) the semi-intensive system of Cornu aspersum production dominates (on average 30 tonnes/ha). In this system 50 % of field breeding pens are covered with wooden pallets on which dry feed for snails is poured, and the bottom of these pallets is used as microshelters for snails. During the autumn harvest, all sown fodder vegetation is already eaten by snails. Mature snails are therefore easily picking up in autumn and then hibernate in air-conditioned hibernation chambers, with losses of 10-15 %. The research carried out by the authors concerns the technology of extensive farming, without feeding animals,where snails usually winter in their field pens under an appropriate frost-resistens fabric. In this context, the introduction of the described microshelters to the Italian technology with lush vegetation remaining in autumn indeed, as the authors have shown significantly improves picking-up, and wintering indoors contributes to the reduction of individual losses in spring. The work is also correct in formal terms and suitable for publishing after minor corrections below.

R1. We sincerely appreciate the reviewer's thorough evaluation and insightful feedback. We extend our deepest gratitude for recognizing the formal correctness and overall suitability of our study for publication. We also sincerely thank you for your positive comments regarding our research. Your encouraging words reinforce the significance of our work in the field of heliciculture and motivate us to continue contributing to this area of study.

Q2. References should be arranged in alphabetical order;

R2. We checked again the requirements for formating the references , and they should be arranged in the order of their citation in the text; idem to the formating style in the initial version of the manuscris.

Q3. Please, but only if possible, suplement the work with information on the improvement of economic parameters caused by the modification of the tested technology. For example, by how many hours the working time was shortened during picking-up, by how much the material losses were reduced with lower mortality, etc.

R3: Thank you for your suggestion. Unfortunately, we did not have access to specific data on the improvement of economic parameters, such as the reduction in working hours during picking-up or the decrease in material losses due to lower mortality. However, we have provided as much economic data as possible based on the available information to support our findings.

Round 2

Reviewer 1 Report

Comments and Suggestions for Authors

Dear authors,

Your research produces findings that can assist farmers in enhancing their snail farming practices.

Authors have considered all the comments and provided responses to each one. The paper is ready for publication as it stands.